# Better Optimization Can Reduce Sample Complexity: Active Semi-Supervised Learning via Convergence Rate Control

## Abstract

Reducing the sample complexity associated with deep learning (DL) remains one of the most important problems in both theory and practice since its advent. Semi-supervised learning (SSL) tackles this task by leveraging unlabeled instances which are usually more accessible than their labeled counterparts. Active learning (AL) directly seeks to reduce the sample complexity by training a classification network and querying unlabeled instances to be annotated by a human-in-the-loop. Under relatively strict settings, it has been shown that both SSL and AL can theoretically achieve the same performance of fully-supervised learning (SL) using far less labeled samples. While empirical works have shown that SSL can attain this benefit in practice, DL-based AL algorithms have yet to show their success to the extent achieved by SSL. Given the accessible pool of unlabeled instances in pool-based AL, we argue that the annotation efficiency brought by AL algorithms that seek diversity on labeled samples can be improved upon when using SSL as the training scheme. Equipped with a few theoretical insights, we designed an AL algorithm that rather focuses on controlling the convergence rate of a classification network by actively querying instances to improve the rate of convergence upon inclusion to the labeled set. We name this AL scheme convergence rate control (CRC), and our experiments show that a deep neural network trained using a combination of CRC and a recently proposed SSL algorithm can quickly achieve high performance using far less labeled samples than SL. In contrast to a few works combining independently developed AL and SSL (ASSL) algorithms, our method is a natural fit to ASSL, and we hope our work can catalyze research combining AL and SSL as opposed to an exclusion of either.

## 1 Introduction

The data-hungry nature of supervised deep learning (DL) algorithms has spurred interest in active learning (AL), where a model can interact with a dedicated annotator and request unlabeled instances to be labeled. In the pool-based AL setting, a model initially has access to a set of unlabeled samples and can query instances which need be labeled for training. Under certain conditions on the task, AL can provably achieve up to exponential improvement in sample complexity and thus has great potential for reducing the number of labeled instances required to achieve high accuracy. This is especially important when the annotation task is extremely costly, for example, in medical imaging where only highly-specialized experts can diagnose a subject's condition.

Active learning algorithms have been extensively explored, with various formulations including uncertainty-based sampling (Wang & Shang, 2014), aligning the labeled and unlabeled distributions (Gissin & Shalev-Shwartz, 2019) with connections to domain adaptation (Ben-David et al., 2010), and coreset (Sener & Savarese, 2018). Furthermore, there is no standard method in modeling a deep neural network's (DNN) uncertainty, and uncertainty-based AL has its own variants ranging from utilizing Bayesian networks (Kirsch et al., 2019) to using a model's predictive confidence (Wang & Shang, 2014). This ambiguous characterization of how much information a sample's label carries also motivated AL algorithms based on maximizing the expected change of a classification model (Huang et al., 2016; Ash et al., 2020).

---

**Algorithm 1** Active Semi-Supervised Learning

   **for** each query iteration $i = 1, ..., N$ **do**
      Train a classifier $f_\theta$ using some SSL algorithm on $(\mathcal{X}_L, \mathcal{X}_U)$ until convergence.
      Retrieve unlabeled samples $\mathcal{X}_u^* \subset \mathcal{X}_U$ using Alg. 2 and obtain their labels.
      Update labeled and unlabeled pools $\mathcal{X}_L \leftarrow \mathcal{X}_L \cup \mathcal{X}_u^*, \mathcal{X}_U = \mathcal{X}_U \backslash \mathcal{X}_u^*$.
   **end for**

---

While AL comes with optimistic potentials, most algorithms outperform random sampling (passive learning) by only a small margin, with follow-up works (Gissin & Shalev-Shwartz, 2019; Sener & Savarese, 2018; Ducoffe & Precioso, 2018) reporting worse performance of certain AL algorithms than random sampling due to their dependency on specific model architectures or dataset characteristics. Furthermore, the performance of AL algorithms are usually reported by training a model using supervised learning (SL) on the queried labeled data despite the availability of unlabeled data in pool-based AL. Semi-supervised learning (SSL) has recently shown impressive performance with a small number of labeled instances, and its most premature variant known as pseudo-labeling (Lee, 2013) has been combined with AL algorithms (Wang et al., 2017). One recent work (Song et al., 2019) uses a rather modern SSL algorithm (Berthelot et al., 2019) and shows the strength of combining AL with SSL which we name ASSL. However, their AL algorithm is not designed specifically considering the ASSL setting.

In this work, we propose a novel query strategy which naturally blends in with SSL, and show that our AL algorithm can rapidly achieve the high performance of fully-supervised algorithms using fewer labeled data. Our algorithm is inspired by recent developments in DNN theory, namely the neural tangent kernel (NTK) (Jacot et al., 2018). Experimental comparisons with diversity-seeking strategies and an algorithm with an objective similar to ours demonstrate how labeling instances based on our objective helps SSL attain high performance in a sample-efficient manner.

## 2 BACKGROUND THEORY

### 2.1 SAMPLE EFFICIENCIES OF ACTIVE LEARNING AND SEMI-SUPERVISED LEARNING

Here we informally describe some theoretical results describing the superiority of AL to passive learning and SSL to SL in terms of labeled sample complexity, that is, the number of labeled instances that can be used to attain $\epsilon$-classification error. Because pool-based AL subsumes unlabeled instances, it makes much more sense to perform SSL on the readily available unlabeled instances when training a classification network between each query iteration (Alg. 1). This section describes how AL and SSL algorithms can consider different objectives to attain higher accuracy in the ASSL setting.

It is well known that SL can find an $\epsilon$-optimal classifier from a sufficiently rich class of hypotheses (e.g. classifiers which can be realized by deep neural networks) using $\tilde{\Theta}\left(\frac{1}{\epsilon}\right)$ i.i.d samples for the separable case and $\tilde{\Theta}\left(\frac{1}{\epsilon^2}\right)$ in the agnostic case (Massart & Nédélec, 2006; Vapnik & Izmailov, 2015)[1]. In contrast, actively adding instances to the (labeled) training set can sometimes improve the sample complexity sub-exponentially from $O(1/\epsilon)$ to $O(\text{poly} \log 1/\epsilon)$ (Balcan et al., 2010). When unlabeled instances are drawn i.i.d., Balcan & Urner (2016) showed how labeling samples selected via binary search over $\tilde{O}(1/\epsilon)$ unlabeled instances can achieve exponential improvement over passive learning.

Göpfert et al. (2019) constructed a few examples which show how an SSL algorithm can use unlabeled instances to significantly improve the labeled sample complexity for a rather restricted class of data distributions. A corollary of one such example is that using $O(\log 1/\epsilon)$ labeled samples and $O(1/\epsilon^2)$ unlabeled samples can be used to obtain $\epsilon$-error. Considering a target error $\epsilon \approx 6\%$ achieved by fully-supervised learning in CIFAR10 using $O(1/\epsilon) \approx 50,000$ labeled samples, a training set with only $O(\text{poly} \log(1/\epsilon)) \approx 250$ labeled samples and leaving the remaining $O(1/\epsilon) \approx 49,750$ samples unlabeled is analogous to the aforementioned sample complexities achievable in the respective AL and SSL settings, although their assumptions may not be satisfied.

Instead of focusing on the same objective for both AL and SSL to attain the potential exponential sample complexity improvements above, we suggest using an AL algorithm that queries instances to

---

[1]Here we slightly abuse the $\tilde{\Theta}$ notation to denote upper and lower bounds matching up to logarithmic factors.

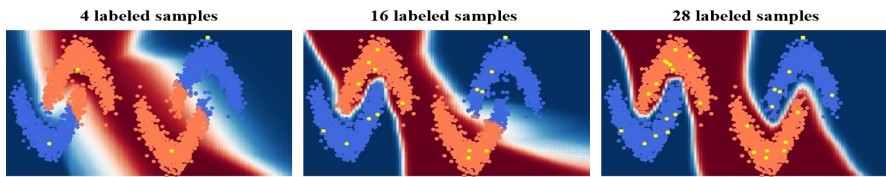

Figure 1: Illustration of how a Π-model's (Laine & Aila, 2017) decision boundary changes as the proposed algorithm enlarges the labeled set. The 4 initial pivot points were manually selected.

help optimization. Specifically, we would like an AL algorithm to construct a labeled dataset such that a classifier $f^{(T)}$ trained for $T$ iterations rapidly minimizes the optimization error $|f^{(T)} - f^{(\infty)}|$, and formulate an SSL objective $\mathcal{L}_n(f)$ to guide the target classifier $f^{(\infty)} \in \arg\min_f \mathcal{L}_n(f)$ towards a solution that minimizes $|f^{(\infty)} - f^*|$, where $f^* \in \arg\min_f \mathbb{E}[\mathcal{L}_n(f)]$ is the optimal classifier and $\mathcal{L}_n$ is the empirical loss computed over $n$ samples. This perspective spans an online setting where less training iterations implies better generalization, i.e. $f^{(\infty)} = f^*$, and AL focusing on $|f^{(T)} - f^{(\infty)}|$ effectively minimize the optimization-generalization error $|f^{(T)} - f^*|$. Together, the resulting model would ultimately minimize the generalization error's upper bound: $|f^{(T)} - f^*| \leq |f^{(T)} - f^{(\infty)}| + |f^{(\infty)} - f^*|$. The next section shows how we can carefully design labeled and unlabeled sets such that a classifier trained on this construction enjoys a fast rate of convergence by employing a recently developed tool known as the NTK.

## 2.2 NEURAL TANGENT KERNEL

The neural tangent kernel (NTK) (Jacot et al., 2018) is a theoretical tool developed to understand how overparametrized networks can be well-optimized and generalize to unseen data. In particular, the training dynamics of DNNs parameterized by $\theta_t$ at any iteration $t$ follows a closed form expression

$$\frac{df_{\theta_t}(\mathcal{X})}{dt} = -\hat{\mathcal{K}}^{(t)}(\mathcal{X}, \mathcal{X})\left(f_{\theta_t}(\mathcal{X}) - \mathcal{Y}\right). \tag{1}$$

when minimizing the mean-squared error (MSE) loss using gradient descent, where $\hat{\mathcal{K}}^{(t)}(\mathcal{X}, \mathcal{X}) = \left[\nabla_\theta f_{\theta_t}(x)^T \nabla_\theta f_{\theta_t}(x')\right]_{(x,x')\in\mathcal{X}\times\mathcal{X}}$ is the empirical NTK (Arora et al., 2019), where each element $\nabla_\theta f_{\theta_t}(x)^T \nabla_\theta f_{\theta_t}(x') \in \mathbb{R}^{C\times C}$ is a matrix block and $C$ is the number of classes. If we let the width of all layers grow to infinity, a remarkable property holds in which the training dynamics can be characterized using a matrix independent of time, namely the NTK $\mathcal{K}$:

$$\frac{df_{\theta_t}(\mathcal{X})}{dt} = -\mathcal{K}(\mathcal{X}, \mathcal{X})\left(f_{\theta_t}(\mathcal{X}) - \mathcal{Y}\right) \tag{2}$$

with probability 1 over random initialization[2]. For any network of finite width $d$ trained for $t$ iterations, the relation (2) holds with probability $\geq 1 - \delta_d$, where $\delta_d \to 0$ as $d \to \infty$.

Assuming a network that satisfies the differential equation, the transient solution is given by

$$f_{\theta_t}(\mathcal{X}) = \sum_i c_i v_i \exp\left(-\lambda_i(\mathcal{X})t\right) \tag{3}$$

for some constants $c_i$ and eigenvalue/vector pairs $(\lambda_i, v_i)$ of the NTK $\mathcal{K}(\mathcal{X}, \mathcal{X})$. Notice how the NTK, and in turn the convergence rate, depends on the set of instances $\mathcal{X}$ it is computed over. This sparks the intuition that the rate of convergence can be *controlled* by carefully designing a labeled set.

## 3 ACTIVE LEARNING VIA CONVERGENCE RATE CONTROL

### 3.1 CONVERGENCE RATE CONTROL

Inspired by the aforesaid NTK's characterization of the model's training dynamics when supervised on the dataset it is computed over, we propose to move unlabeled instances $\mathcal{X}_u^*$ selected from a

---

[2]Formally, the dynamics $\frac{df_{\theta_t}(\mathcal{X})}{dt}$ converges to the RHS in probability as the network's width sequentially grows to infinity when initialized using He initialization.

pool of unlabeled samples $\mathcal{X}_U$ that maximize the rate of convergence of the SGD dynamics to the labeled set $\mathcal{X}_L$, hence the name convergence rate control (CRC). While Eq. 3 is a solution to the time-inhomogeneous ODE in Eq. 2, the performance of infinitely-wide networks fall behind their finite-width counterparts. We thus replace the eigen-pair $(\lambda, v)$ of the true NTK $\mathcal{K}$ with its empirical NTK $\hat{\mathcal{K}}^{(T)}$ in Eq. 1 using a model trained for $T$ epochs until its validation accuracy saturates.

Maximizing the eigenvalues of the NTK is a multi-objective problem, and CRC takes a worst-case approach to select unlabeled instances that maximize the smallest eigenvalue of the NTK:

$$\max_{\mathcal{X}_u \subset \mathcal{X}_U} \min_{i \in |\mathcal{X}_L \cup \mathcal{X}_u|} \lambda_i \left( \hat{\mathcal{K}}^{(T)} \left( \mathcal{X}_u \cup \mathcal{X}_L, \mathcal{X}_u \cup \mathcal{X}_L \right) \right). \tag{4}$$

Each block $\hat{\mathcal{K}}^{(T)}(x, x') \in \mathbb{R}^{C \times C}$ is replaced by its trace computed as $\sum_c \hat{\mathcal{K}}_{cc}^{(T)}(x, x')$, which we numerically found to be an accurate replacement to the block matrices in computing the minimum eigenvalue, to reduce both computational and memory complexities. In general, the eigenvalue distribution is not necessarily supported over positive values, and we observed that considering only positive eigenvalues consistently results in higher performance than when considering both negative and positive eigenvalues. CRC thus scores unlabeled instances as the minimum over positive eigenvalues. The solution to Eq. 4 can then be found using dedicated GPU-implementations available in popular DL frameworks such as PyTorch. This algorithm is detailed in Alg. 2.

A labeled sample's influence on a DNN's predictive behavior is abstruse. Intuitively, a labeled sample may be most informative when it is near a region separating different classes; however, this does not guarantee that a DNN learns such a decision boundary effectively. Furthermore, it is unclear whether a sample in some other region could better enhance a classifier's performance when added to the labeled set, especially when considering SSL algorithms which complicate DL analysis. Contrary to the above intuition, Rebuffi et al. (2019) demonstrates how a DNN trained with a SSL objective (Laine & Aila, 2017) learns a decision boundary using only 4 manually selected pivot points not on the class-separating region. Figure 1 illustrates how the classifier's decision boundary progressively forms near low-density regions as CRC enlarges the labeled set.

## 3.2 Theoretical Analysis

This section theoretically analyzes the effect of estimating the dynamics using the empirical NTK to construct a labeled set for least squares regression $\ell(\theta) = \frac{1}{2} \|f_\theta(\mathcal{X}) - \mathcal{Y}\|_2^2$. We begin with a deviation bound of the empirical NTK from (Arora et al., 2019).

**Theorem 1** (Paraphrased Theorem 3.1 in (Arora et al., 2019)). *Consider a fully-connected network with ReLU nonlinearities of width $d$ and depth $L$. At initialization, taking the form specified in (Arora et al., 2019), the following holds with probability $\geq 1 - \delta$*

$$\max_{(x_i, x_j): \|x_i\|, \|x_j\| \leq 1} |\hat{\mathcal{K}}^{(0)}(x_i, x_j) - \mathcal{K}(x_i, x_j)| \leq \frac{L^{7/2}}{d^{1/4}} \log \frac{L}{\delta} =: \epsilon.$$

The above statement shows a general case in which the empirical NTK can be expressed as a linear combination of the true NTK and a noise matrix parametrized by $\epsilon > 0$, as long as the empirical NTK does not change abruptly. A similar result holds for ResNets and CNNs given appropriate conditions (Lipschitz, smooth, non-polynomial, analytic) on non-linearities and initialization (Du et al., 2019), where it has also been shown that $\hat{\mathcal{K}}^{(t)}$ provably stays close to $\mathcal{K}^{(0)}$ throughout training. The next proposition gives a tractable approximation of the true NTK's (unknown) eigenvalue as a neighborhood of the empirical NTK's (known) eigenvalue.

**Proposition 1** (CRC approximation error due to empirical NTK). *Suppose $\mathcal{K} = \hat{\mathcal{K}}^{(t)} + \mathcal{N}^{(t)}$ with a symmetric noise matrix of bounded spectral norm $\rho(\mathcal{N}^{(t)}) < \epsilon$. Then, the true NTK's eigenvalues lies in an $\epsilon$-neighborhood of the empirical NTK's eigenvalues with respect to $\|\cdot\|_\infty$, i.e. $\lambda(\mathcal{K}) \in \mathcal{B}_\epsilon \left( \lambda \left( \hat{\mathcal{K}}^{(t)} \right) \right)$, where the eigenvalues $\lambda(\cdot)$ are listed monotonically.*

This statement has nice implications on optimizing DNNs. Consider Theorem 5.1 in (Du et al., 2019) that states running gradient descent with learning rate $\eta_t = O\left( \frac{\lambda_{\min}(\mathcal{K}(\mathcal{X}_L, \mathcal{X}_L))L^2}{|\mathcal{X}_L|^2} \right)$ on least-squares regression using a fully-connected network of sufficient width converges linearly at rate $1 - \frac{\eta \lambda_{\min}(\mathcal{K}(\mathcal{X}_L, \mathcal{X}_L))}{2}$. Two remarks follow:

**Remark 1** (Better conditioning of learning rates). *The sufficient condition on learning rate is proportional to the (unknown) minimum eigenvalue of the NTK. This implies that the increased minimum eigenvalue allows for a larger set of admissible learning rate schedules. In (Athiwaratkun et al., 2019), it has been shown that SSL algorithms take larger steps throughout training, continually exploring parameters. Since CRC permits larger learning rates, tuning the step-size may be easier and the larger steps taken by SSL algorithms can still converge when the step-size is over-estimated.*

**Remark 2** (Direct estimate of sufficient learning rate and convergence rate guarantees). *Both the admissable learning rates and convergence rate are unknown. Invoking Proposition 1 after selecting $\mathcal{X}_L$ using CRC with $\hat{\mathcal{K}}^{(T)}$, we can set $\eta = O\left(\frac{\lambda_{\min}(\hat{\mathcal{K}}^{(T)}(\mathcal{X}_L, \mathcal{X}_L))L^2}{|\mathcal{X}_L|^2}\right)$ which would guarantee linear convergence with the rate deviating at most by the spectral norm $\epsilon$ of the noise matrix.*

### 3.3 PRACTICAL CONSIDERATIONS

Summing the network's gradients over all layers for each instance pair is infeasible for modern network architectures, and we compare one realization of using all parameters with the final layer's to compute the NTK. A comparison of one realization using the NTK computed over all layers vs. using just the final layer averaged over 3 trials (Tab. 1) shows how using the final layer performs just as well as when all layers were used but with much faster ($\times 100$) queries. If we assume that the shallow layers saturate more quickly than deeper layers, justified by the 'lower layers learn first' phenomenon (Raghu et al., 2017), the NTK estimate computed at the network's last layer is a prediction of the training dynamics corresponding to a single layer perceptron attached to a (slowly-changing) feature extractor. In this perspective, the final accuracy is mostly affected by the final layer which determines how high-level features are recognized, and designing a labeled set using the NTK estimated over the last layer would sufficiently capture the model's training dynamics.

| Initial Pool | Randomly initialized model | | | Random sampling | | |
|:---:|:---:|:---:|:---:|:---:|:---:|:---:|
| # Labeled | 50 | 100 | 150 | 100 | 150 | 200 |
| Last | 53.12±5.50 | 67.14±4.28 | 71.98±4.04 | 78.41±3.16 | 83.67±3.24 | 85.67±1.31 |
| All | 50.47 | 68.25 | 74.03 | 71.26±3.92 | 78.13±4.32 | 79.10±3.64 |

Table 1: CRC performances when NTK is computed over final vs. all layers with $G = 5$ at $Q = 50$. Left describes performances when initial query and initial SSL training is performed with a randomly initialized model (one trial for 'All'); right uses random sampling for the first query.

One reason why estimating the training dynamics over all layers could be *worse* than using the last is that the former requires a network trained in the next SSL phase to learn a similar low-level representation to that obtained in the previous SSL phase. However, it was observed by Raghu et al. (2017) that two randomly initialized networks trained on the same dataset have similar final layers but extremely different intermediate representations. Because our networks are retrained from scratch between each query iteration, it is highly likely that the next network will have very different intermediate representations, and these middle layers would provide noisy estimates as to how much the model's functional representation would be affected by the enlarged dataset. Since we are seeking to modify a dataset in order to amplify a model's performance, it is reasonable that we enlarge the labeled set based on how much the additional samples would affect the model's final layers. Had we designed the experiments such that a model is continuously trained from its previous SSL phase, computing the NTK over all layers may have been beneficial at the expense of computation.

### 3.4 MYOPIC VS. BATCH-MODE ACTIVE LEARNING

Myopic AL policies refer to algorithms that score unlabeled instances based on their individual importance. Considering how an expert annotator 'in-the-loop' must label queried samples, it is desirable that an AL algorithm queries large batches based on their collective importance. Myopic policies usually degrade in performance as the query size $Q$ increases, as they may repetitively sample identical instances (Kirsch et al., 2019), and it has been of great interest in designing a computationally efficient batch-AL algorithm. CRC (Alg. 2) is a batch-mode AL algorithm as it computes $\lambda_{\min}$ over groups of unlabeled instances $\mathcal{X}_u$. To cope with the combinatorial complexity associated with the outer-maximization in Eq. 4, we randomly sample candidate groups $\mathcal{X}_u$ without

---

**Algorithm 2** Convergence Rate Control (batch-mode solution to equation 4)

---

Inputs: Unlabeled pool $\mathcal{X}_U$, query-size $Q$, group-size $G$.
Output: New pool $\mathcal{X}_U^*$ to be labeled.
**for** $t = 1, \cdots |\mathcal{X}_U|/G$ **do**
    $\mathcal{X}_u^{(t)} \leftarrow G$ unlabeled instances randomly sampled from $\mathcal{X}_U$.
    $s(\mathcal{X}_u^{(t)}) \leftarrow \lambda_{\min}^+ \left( \mathcal{X}_L \cup \mathcal{X}_u^{(t)} \right)$ using the empirical NTK computed over final layers.
**end for**
Store $Q/G$ highest groups: $t_1^*, \ldots t_{Q/G}^* \leftarrow \arg\max (s)$.
**return** $\mathcal{X}_u^* \leftarrow \mathcal{X}_u^{(t_1^*)} \cup \cdots \cup \mathcal{X}_u^{(t_{Q/G}^*)}$

---

replacement. This randomization significantly reduces the search space, potentially yielding highly sub-optimal solutions, and we introduce a hyperparameter $G$ such that $G$ divides $Q$, and search over candidate groups $|\mathcal{X}_u| = G$, selecting $Q/G$ groups at each query step. The group size controls whether the outer maximization is solved more accurately ($G = 1$) without considering the collective significance of a group, or the inner-minimization incorporates the collective importance of all queried samples ($G = Q$) at the expense of a substantially smaller search space. This trade-off is established in Tab. 2 when $Q = 100$, averaged over three realizations for $G = 2, 5, 10, 100$. Our relatively small group-sizes make the cubic complexity of the inner-minimization negligible, and a larger group size is consequently faster due to less ($Q \rightarrow Q/G$) eigen-decomposition operations and inefficient batch-gradient computations in autograd packages. All experiments hereon use $G = Q/10$ considering the speed vs. performance trade-off.

| Query step | 1 | 2 | 3 | 4 | 5 |
|---|---|---|---|---|---|
| $G = 2$ | $84.72 \pm 0.96$ | $87.11 \pm 1.14$ | $87.83 \pm 0.17$ | $88.3 \pm 0.30$ | $89.44 \pm 0.59$ |
| $G = 5$ | $82.55 \pm 3.21$ | $84.13 \pm 2.64$ | $87.24 \pm 2.42$ | $87.66 \pm 1.33$ | $88.63 \pm 0.60$ |
| $G = 10$ | $83.38 \pm 1.14$ | $85.71 \pm 1.50$ | $88.07 \pm 0.06$ | $88.36 \pm 0.71$ | $89.35 \pm 0.79$ |
| $G = 100$ | $83.31 \pm 2.17$ | $86.49 \pm 1.47$ | $86.76 \pm 2.56$ | $88.49 \pm 1.43$ | $88.63 \pm 1.67$ |

Table 2: Effect of different group sizes with $Q = 100$ using random sampling for the fist query.

### 3.5 RELATION TO GRADIENT-BASED METHODS

The proposed algorithm can be interpreted as measuring the importance of unlabeled instances by how much they would change a model's parameters. Two AL algorithms are similar in this manner in that they seek to label instances which affect the model the most: expected gradient length (EGL) (Huang et al., 2016) and BADGE (Ash et al., 2020). EGL first computes the gradient of the cross-entropy loss over all unlabeled instances, replacing the ground truth target with an expectation over a posterior distribution determined by the trained model, then selects instances with the greatest norm. BADGE uses an embedding of the loss' gradients using predicted classes, computed over the parameters in just the final layer, and uses kmeans++ to enforce diversity. Nonetheless, both algorithms only estimate the benefit due to candidate unlabeled samples and do not measure the collective influence of the labeled set.

## 4 EXPERIMENTS

### 4.1 IMPLEMENTATION DETAILS

Our SSL implementation is primarily based on different publicly-available PyTorch implementations of FixMatch (Sohn et al., 2020) for CIFAR10[3] and CIFAR100[4], as well as mean teacher (Tarvainen & Valpola, 2017) for SVHN[5] that performed best at the time of this work. We made modifications as

---

[3]https://github.com/kekmodel/FixMatch-pytorch
[4]https://github.com/LeeDoYup/FixMatch-pytorch
[5]https://github.com/perrying/realistic-ssl-evaluation-pytorch

| Accuracy (%) | Q=50 | | | | | Q=100 | | | | |
|---|---|---|---|---|---|---|---|---|---|---|
| # Labeled | 100 | 150 | 200 | 250 | 300 | 200 | 300 | 400 | 500 | 600 |
| Random | 69.81±4.77 | 82.14 ±2.30 | 82.41±0.68 | 84.98±2.32 | 84.92±0.73 | 85.04±1.53 | **87.09**±1.25 | 87.57±0.57 | 87.92±0.47 | 87.37±2.18 |
| Entropy | 72.43±8.21 | 82.41±2.89 | 83.38±1.93 | 84.86±1.83 | 86.42±0.95 | **85.51**±1.47 | 86.89±0.47 | 86.87±1.22 | 88.27±0.30 | 88.44±1.17 |
| Confidence | 52.54±7.71 | 49.55±7.77 | 50.67±3.15 | 45.04±8.25 | 53.44±8.75 | 77.05±4.66 | 78.29±5.55 | 73.68±5.12 | 73.69±2.50 | 75.98±3.29 |
| EGL | 57.18±7.41 | 56.67±10.53 | 68.76±12.57 | 74.47±12.32 | 76.67±7.43 | 79.95±3.11 | 80.41±2.77 | 81.20±5.10 | 82.62±0.36 | 83.55±3.36 |
| CRC† | **78.41**±3.16 | **83.67**±3.24 | **85.67**±1.31 | **86.88**±0.87 | **87.01**±0.28 | 83.38±1.14 | 85.71±1.50 | **88.08**±0.06 | **88.37**±0.71 | **89.35**±0.79 |

Table 3: Performances on CIFAR10 using $Q \in \{50, 100\}$ with random sampling for the first query.

suggested in (Oliver et al., 2018). In particular, we used the WRN-28-2 and WRN-28-10 architectures for SVHN/CIFAR10 and CIFAR100, respectively, further motivated by the derivation of NTK as a network in its infinite width limit. Mean teacher (MT) enforces consistency between the model trained using SGD and an exponential moving average (teacher) of the model's weights. FixMatch is a variant of MixMatch (Berthelot et al., 2019), replacing standard augmentations with more powerful augmentation techniques when computing the SSL loss. In short, FixMatch enforces consistency between a classification model's predictions on perturbed inputs which effectively yields a more robust model with small Jacobian norm (Athiwaratkun et al., 2019) in junction with other SSL algorithms including MT. Furthermore, the hyperparameters in FixMatch were kept the same across different dataset sizes in their original report, making our experimental comparisons convenient.

The initial pool of labeled instances significantly affected the downstream SSL task and subsequent AL queries, especially for uncertainty-based algorithms (see section 4.2). It is practical to assume that a model would initially have access to a small pool of labeled images with an equal number of classes. Thus, we randomly sampled a balanced number of images per class in the first query iteration and use the AL algorithms in subsequent queries without enforcing balanced classes. Between each query step, the classification network was trained until either its validation accuracy does not improve for $50/100/\infty$ epochs or $350/600/512$ epochs is reached for CIFAR10/100/SVHN. All experiments are repeated 3 times with mean ± std performances reported unless stated otherwise. Our implementation will be open-sourced upon publication.

## 4.2 BASELINE ALGORITHMS

Both AL and SSL algorithms demand much more time than standard supervised learning, and an extensive comparison of all ASSL combinations is beyond our compute availability. We resort to comparing the proposed algorithm with 3 other AL algorithms, 2 of which scores unlabeled instances based on how much they reduce a model's uncertainty, and the other because of its similarity to CRC. Uncertainty-based AL (Wang & Shang, 2014) is one of the simplest DL-based AL algorithm that selects the unlabeled instance with lowest confidence in its prediction: $\mathcal{X}_u^* = \arg\min_{\{x \in \mathcal{X}_U\}} \max_y f_{\theta,y}(x)$. In the same work, Wang & Shang (2014) proposed to also use entropy as a measure of the model's uncertainty in unlabeled samples. EGL (Huang et al., 2016) retrieves instances that modify a model the most: $\mathcal{X}_u^* = \arg\max_{\{x \in \mathcal{X}_U\}} \mathbb{E}_{y \sim \hat{p}(\cdot;x,\theta)} \left[ \|\nabla_\theta H(y, f_\theta(x))\|^2 \right]$ where $\hat{p}(y; x_i, \theta) = f_{\theta,y}(x_i)$ is the model's confidence in class $y$ and $H$ is the cross entropy function.

## 4.3 ACTIVE SEMI-SUPERVISED LEARNING

Here we report the sample efficiencies obtained by different SSL algorithms when the labeled set is constructed using various AL schemes in Tabs. 3 and 4. For reference, the implementations used for each experiment achieve average accuracies of SVHN (100/class): 93.52%, CIFAR10 (25/class): 87.33, and CIFAR100 (40/class): 48.96% with balanced sampling. Note, however, that SSL settings consider labeled sets (and consequently, unlabeled sets) with *equal numbers of images per class*, putting SSL settings to an unrealistic advantage at the same number of labeled images. Different levels of class-imbalance result in larger performance variance, which is especially critical in the low-data regime.

One difficulty arising from the ASSL setting is class imbalance. In contrast to standard AL settings that consider much larger query sizes, the small query size in ASSL significantly impacts the class distribution, resulting in occasional performance drop for certain query strategies. Entropy has repeatedly achieved state-of-the-art performance (Gissin & Shalev-Shwartz, 2019; Wang et al., 2017; Ash et al., 2020) in AL settings despite its simplicity, and our experiments show the same holds for ASSL. Conforming with our motivation in section 2.1, CRC performed comparable or better

| Accuracy (%) | SVHN | | | | | CIFAR-100 | | | |
|---|---|---|---|---|---|---|---|---|---|
| # Labeled | 140 | 160 | 180 | 200 | 220 | 250 | 300 | 350 | 400 |
| Random | **56.03**±5.88 | **66.48**±12.53 | 73.46±1.73 | 89.20±1.08 | 91.01±0.95 | 41.45±1.04 | 44.57±0.74 | 45.19±0.91 | 49.35±1.55 |
| CRC† | 55.01±3.98 | 64.02±4.58 | **85.14**±5.01 | **89.89**±0.63 | **91.44**±0.23 | **41.87**±1.45 | **45.27**±0.59 | **47.04**±0.50 | **50.16**±0.37 |

Table 4: Mean teacher (SVHN: $Q = 20$) and FixMatch (CIFAR100: $Q = 50$) performances with 12 and 20 randomly sampled images per class in the first query.

than entropy although it does not explicitly enforce diversity on the labeled set. Furthermore, while both CRC and EGL scores instances by estimating how much they would affect the model's training dynamics, only CRC outperformed random sampling. Our results suggest that actively enlarging the labeled set can definitively reduce the labeled sample complexity in the SSL setting, without having to construct a labeled set consisting of images representative of their class.

Existing DL-based AL algorithms have performed comparably to random sampling when using large query sizes, and our experiments show the same for entropy and CRC with $Q = 100$ in the ASSL setting. A proper AL algorithm should achieve higher performance at the same number of labeled samples if the query size is small since it would be taking more query steps in order to claim its superiority to random sampling; however, we observed that only CRC satisfies this property. EGL's low performance had been reported on image classification tasks (Gissin & Shalev-Shwartz, 2019; Sener & Savarese, 2018; Ducoffe & Precioso, 2018) under the AL setting. Our experimental design of retraining networks from scratch may have had negative influence on EGL, similar to how it affected the NTK estimate over all layers. We attribute confidence's low performance to the poor calibration of DNNs (Guo et al., 2017), inadequately capturing the model's uncertainty on unlabeled instances. A successful application of confidence-based query may require incorporating calibration while training classification networks or before performing such queries.

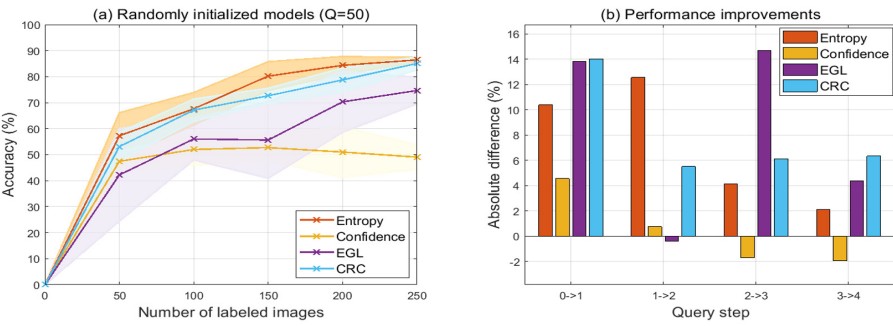

Figure 2: Performances when the initial AL queries were performed using a randomly initialized model. The figure on the right underscores performance gains in consecutive query steps.

Ideally, an AL algorithm should have a nice transferrability property, constructing labeled sets that are useful to train other networks. This obviously holds for random sampling, as the queries do not depend on the model's weights. To validate the transferrability of dataset constructions of AL algorithms, we used a randomly initialized model to query only the initial pool, trained the model using a different realization of initialization, and as done previously for other experiments, queried additional labels using the model trained using SSL. As shown in Fig. 2, however, such initial queries adversely affected all subsequent queries. While entropy and CRC overcome this initial disadvantage after a few query iterations, other baselines tend to under-perform random sampling due to this initial disadvantage. This phenomenon has not been reported in previous works, and whether this is an artifact of our ASSL setting or the architecture/hyperparameters used deserves future notice.

### 4.4 CONVERGENCE RATE AND TIME HORIZON PREDICTION

The proposed CRC algorithm is designed to query unlabeled instances that improve the rate of convergence a DNN in the following SSL phase. To see if our algorithm achieves this purpose, we plot the empirical NTK's minimum eigenvalues used to score unlabeled instance groups against the number of training epochs until the test *loss* reaches its global minimum in Fig. 3 (a) to show how the

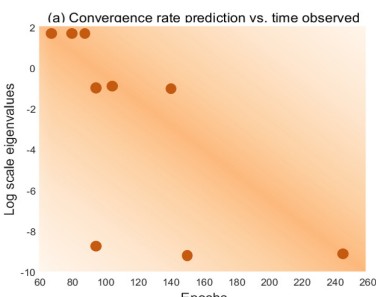
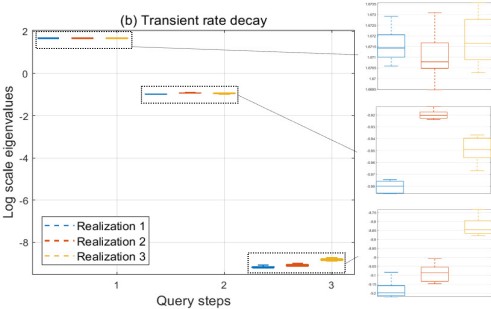

Figure 3: Minimum eigenvalues' distribution in log-scale ($Q = 50$). Left figure shows how informative the minimum eigenvalues are in terms of predicting the number of epochs required for convergence, and right figure illustrates how the minimum eigenvalues' distribution concentrates towards zero. Negative linear slope implies (a) exponential convergence as predicted by Eq. 3 and (b) slower convergence rate with increasing labeled set size.

minimum eigenvalues are predictive of the time horizon. Global minimum here refers to the value for which the loss never falls below until training completes. We additionally plot how the minimum eigenvalues' distribution progresses with query steps, where Fig. 3 (b) shows that the eigenvalue distribution concentrates towards 0 as the labeled set grows, implying slower convergence rates. A similar observation was made by Nitanda & Suzuki (2020) where the authors showed how the true NTK's eigenvalues concentrate towards zero as the training set size increases, whereas our plot is over a union of labeled and queried groups. This figure illustrates how all candidate datasets will inevitably have eigenvalue distributions concentrated near zero, but the enlarged dataset chosen by CRC will have the fastest convergence rate among all considered candidates.

## 5 CONCLUSION

This work motivated the combination of AL with SSL, using the former for optimization and latter objective to restrict the hypothesis space to those that generalize well. We then proposed an AL algorithm inspired by the NTK, and demonstrated how controlling the convergence rate of the training dynamics can significantly improve sample efficiency. To the best of our knowledge, this work is the first practical application of NTK outside of supervised learning, where it has been shown that the true NTK is inferior to its DNN counterpart. In contrast to most DL-based AL experiments which use relatively small networks, this work demonstrates that DL can benefit from AL using modern architectures. Our algorithm outperformed uncertainty-based AL algorithms which enforce diversity and an AL algorithm with a similar goal of maximizing the model's change. The proposed algorithm's superiority was also demonstrated in the batch-AL setting, where an optional hyperparameter can be used to control the trade-off between computational complexity and performance.

Our CRC algorithm is developed under the assumption that a well-optimized network $f^{(\infty)}$ induced by the SSL objective is a good approximation of a model $f^*$ that generalizes well. Incorporating the interplay between AL and SSL seems promising in further reducing the sample complexity, designing either an SSL algorithm best fit for some fixed AL algorithm or a joint ASSL scheme. For example, CRC provides an estimate on the number of epochs required for convergence, and a learning rate schedule best fit for the estimated time horizon could be used (Ge et al., 2019).

The proposed CRC algorithm can be extended in several directions, applying generalization guarantees characterized by the NTK to an online (streaming) setting where unlabeled instances cannot be stored. Another exciting direction to pursue would be to use the True NTK (Arora et al., 2019; Novak et al., 2020) to estimate the training dynamics. While CRC is motivated by controlling the transient solution using an estimated proxy of the training dynamics, the true NTK would enable an exact computation of the ODE solution. Of utmost importance would be that this could allow AL queries to be performed offline, that is, without having to train a model between each query iteration, alleviating the long training times associated with ASSL.

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

# A   APPENDIX

## A.1   EXPERIMENTAL DETAILS

All experiments described samples equal numbers of images per class on the first query step. Those described as "using random sampling" uses the Random (passive) AL scheme, i.e. sampling uniformly at random, and "randomly initialized models" refers to transferrability, where the first query step is performed using a randomly initialized model (e.g. confidence of a randomly initialized model when using the "Confidence" AL scheme), then performing SSL with another instance of random initialization. When random sampling was performed, we ran several experiments and used the best models at query step 0, since we observed (similar to transferrability experiments) that the first query step may highly influence subsequent steps, affecting the performance of different AL algorithms resulting in an unfair comparison. This is why all experiments with first query being random sampling (query step 0) excluded the first query step, as they all used identical models when performing the first actual AL query (query step 1). All subsequent queries were performed using the trained model obtained from the previous SSL step, regardless of transferrability. Taking "Confidence" on CIFAR10 as an example for the AL scheme, a model trained using SSL at query step 1 is loaded to query the next 50 (when $Q = 50$) labeled samples for query step 2.

Architectures used for SVHN/CIFAR10/CIFAR100 are WRN-28-2/WRN-28-2/WRN-28-8, where the last is a known requirement due to more classes. SVHN used leaky ReLU with negative slope 0.1 for Mean Teacher, and the others used ReLU activations. "One epoch" is defined as the number of SGD steps between each validation, which we set as 1024 training iterations (far more frequent than the default parameters for Mean Teacher on SVHN). All other configurations were as set in the original implementations.

## A.2   PROOF OF PROPOSITION 1

*Proof.* Let $\mathcal{N}^{(t)} = \mathcal{K} - \hat{\mathcal{K}}^{(t)}$ and denote $\lambda(\cdot)$ the eigenvalues of some matrix of size $n$, listed in non-increasing order. Since $\mathcal{K}$ and $\hat{\mathcal{K}}^{(t)}$ are Hermitian by definition, Weyl's inequality gives

$$\lambda_i\left(\hat{\mathcal{K}}^{(t)}\right) + \lambda_j\left(N^{(t)}\right) \leq \lambda_k\left(\hat{\mathcal{K}}^{(t)} + \mathcal{N}^{(t)}\right) \leq \lambda_r\left(\hat{\mathcal{K}}^{(t)}\right) + \lambda_s\left(N^{(t)}\right), i+j-n \geq k \geq r+s-1.$$

This implies

$$\lambda_k\left(\hat{\mathcal{K}}^{(t)}\right) + \lambda_n\left(\mathcal{N}^{(t)}\right) \leq \lambda_k\left(\hat{\mathcal{K}}^{(t)} + \mathcal{N}^{(t)}\right) \leq \lambda_k(\hat{\mathcal{K}}^{(t)}) + \lambda_1(\mathcal{N}^{(t)}), \forall k \in \{1,\ldots,n\}$$

and hence,

$$|\lambda_k\left(\mathcal{K}\right) - \lambda_k\left(\hat{\mathcal{K}}^{(t)}\right)| = |\lambda_k\left(\hat{\mathcal{K}^{(t)}} + \mathcal{N}^{(t)}\right) - \lambda_k\left(\hat{\mathcal{K}^{(t)}}\right)| \leq \|\mathcal{N}^{(t)}\|_2,$$

where $\|\cdot\|_2$ is the spectral norm. $\qquad\square$

