# OpenReview forum: "Better Optimization can Reduce Sample Complexity: Active Semi-Supervised Learning via Convergence Rate Control"
_ICLR.cc/2021/Conference — Reject_

### Official Review · AnonReviewer3 · 2020-10-26

**Rating:** 5
**Confidence:** 3

**Review:**

Summary: This paper combines semi-supervised learning with active learning, arguing that we should try to focus on actively choosing to label points that improve the convergence rate of the model after adding that example to the training set. They argue that especially in pool-based active learning, where an unlabeled pool of candidate data points are available to choose from to label, methods should also use this unlabeled pool for semi-supervised learning. To optimize convergence rate, they try to select points that maximize the smallest eigenvalue of the empirical NTK over the final layer only, as an approximation (which the authors show seems to do similarly to computing the full NTK, when the next training episode of the active learning is warm-started with the weights of the previous episode.

Strengths:
- The method is general to any SSL method, and the authors consider one of the more recent SSL methods, FixMatch.
- The use of SSL in the pool-based active learning setup makes good sense.
- They use a nice tractable formulation of the convergence rate optimization objective through the eigenvalues of the NTK on the final layer only.
- They show empirically that more positive eigenvalues of the last layer NTK leads to convergence in fewer training epochs.
- The paper suggests a subtle difference between randomly initializing the networks between each phase of active learning and warm-starting from the previous model, noting that the confidence scores, etc. used to select examples for active sampling could be dependent on the random initialization, noise during optimization, among others, and thus may not transfer to the next phase if the model is then randomly re-initialized.

Weaknesses:
- The authors argue that since active learning and semi-supervised learning both have the potential to improve sample efficiency exponentially, their combined benefit is "most likely marginal". However, there is not much theoretical or empirical evidence for this beyond some intuitions. It's quite unclear whether the combined benefit is only marginal, since the assumptions on the data that are needed for semi-supervised learning and active learning are not really the same. This would seem to suggest that combining them still makes sense to robustify the gains in sample complexity, even ignoring potential stacking of the gains.
- The convergence rate control method seems to be motivated by wanting the active learning component to "optimize something different" to the SSL component (which is optimizing for test generalization). It's unclear whether faster convergence rate is really different from improving sample complexity/test generalization. Consider running a learning algorithm on a stream of data examples sampled IID. Consider an algorithm that can learn to low test error with few samples; this means that with few samples, the model can get low loss on unseen samples from the distribution, leading to low loss if you continued optimization. This would mean that the training has more or less converged. Now consider a deep learning algorithm with fast convergence rate; this means that with few iterations (read: few samples seen), the algorithm converges to a solution that continues to have low loss if you continue optimization (and thus low error on IID samples from the same distribution => low test error). Could this method of selecting examples to improve the convergence rate be actually optimizing something similar to improving generalization, instead of doing something alternative?
- In Algorithm 2, it seems like Q/G random groups of unlabeled data are considered, and the Q/G best groups are added to the dataset. Doesn't this mean that all the samples in the Q/G random groups will be added regardless of the NTK eigenvalue calculation?
- Empirically, it's unsatisfying that even with more labeled images than FixMatch (with 250 labeled examples, they get 94\%), this FixMatch + active learning method gets worse accuracy (85\%, Table 3, 300 labeled examples). What is the performance of FixMatch without active learning using the training setup that the authors used, just for comparison? It seems surprising that some of the Fixmatch + active learning methods get only about 50% accuracy when Fixmatch itself can get >90\%. If the labeled seed set were balanced, would we see results surpassing Fixmatch?

Other:
- Is it true that reinitializing the networks hurts CRC but doesn't hurt the Entropy method much? Why is that?

---

> ### Author Response · Authors · 2020-11-19
> **Response**
>
> * Response to comments 1&2: (1) Tone regarding marginal benefit of combining AL with SSL and (2) Faster convergence may not be different from improving sample complexity
>     * The fact that faster optimization and requiring fewer training samples is equivalent in the online setting is an excellent example coinciding with the case that a well-trained model is the optimal f^(infty)=f^*, with the second term |f^(infty)-f^*|=0 in the upper bound described in Section 2.1. Our claim that CRC minimizes the optimization objective f^T - f^(infty)=f^T-f^* agrees with the online setting of minimizing equal to f^T-f^* described by the reviewer.
>     * We acknowledge that section 2.1 is described in a tone that makes it seem that the two objectives should be orthogonal. It is true that two algorithms considering the same objective may robustify their gains, especially when the underlying assumptions differ. The description above integrates both cases of when optimization and generalization overlap significantly (e.g. example given by the reviewer), robustifying the gain, and under settings in which it is better to focus on separate objectives (the upper bound in triangular inequality), effectively minimizing f^T - f^*. We have rephrased Section 2.1 to mitigate this tone of implying orthogonality of objectives is necessary.
> * Algorithm 2
>     * There was a now-corrected typo in the original manuscript. We fixed Alg. 2 to iterate over |X_U|/G groups. Since the top-Q/G groups (each of non-overlapping G elements) are queried, we obtain Q unlabeled samples (of the |X_U| unlabeled samples).
> * Unsatisfactory performance
>     * Please see the overall response to all reviewers for an overview of the reason for performance degradation, where we describe one reason for unsatisfactory performance and the updated manuscript with additional SSL algorithms and different implementations. In addition, class-imbalance due to the same number of images per class not being enforced, as the knowledge of images corresponding to different classes prior to querying would defeat the purpose of AL. Class-imbalance may especially result in lower performance when there are few labeled images; for example class 1 may not be sampled in any subsequent queries (other than query step 0 where class-balance is explicitly enforced) for a particular ASSL realization.
>     *  Performance of FixMatch purely under our setting should be referenced as “Random”.
> * Other: Reinitialization hurts CRC but not Entropy as much
>     * Both entropy and CRC performed better than the other two baselines when the first query was performed using randomly initialized models. This is a phenomenon of "transferrability" (term describing the last paragraph in Section 4.3) of labeled samples, similar to how random sampling does not depend on the model's initialization. Samples queried by AL algorithms which were hurt by the reinitialization are thus biased towards the particular initialization, and are not as useful when training another network (from another realization of initialization).

---

> > ### Comment · AnonReviewer3 · 2020-11-22
> > **Response to authors**
> >
> > Thanks to the authors for the detailed response. On the topic of the motivation for focusing on convergence rate, the new framing of the motivation makes sense - thanks for the clarification. The authors view the AL component as reducing the estimation error of running SGD with T iterations, and the SSL component as reducing approximation error not by changing the function family but by changing the space of minimizers.
> >
> > The main remaining concern was the performance degradation from FixMatch. Considering the class imbalance, it would be good to disentangle the effects of AL in terms of imbalance vs. other sources. For example, the authors claim that due to the active learning setting, it would not make sense to have a balanced seed set which requires looking at some labels. However, one could imagine taking an imbalanced seed set and subsampling to get a balanced seed set to begin with, and then allowing for the unused labels to be chosen during the active learning process without a query cost. In that case, it would be good to see a comparison of FixMatch on the regular balanced labeled set vs. CRC and show that there is improvement in this regime too.
> > Regarding the unfortunate issue in the codebase of FixMatch - perhaps it's worth checking that the results still hold with the changes, but if I understand correctly, the performance difference should more be due to the class imbalance than the code issues.
> >
> > Taking these into account, I upgrade my score to 5. There is still a bit of uncertainty on the experimental side which if cleared up would give me more confidence, especially given that the current results do not over-perform baselines all the time. I'd be glad to increase again if these could be addressed, though I recognize the time constraint.

---

### Official Review · AnonReviewer1 · 2020-10-27
**This paper is interesting but has a large room to improve**

**Rating:** 5
**Confidence:** 3

**Review:**

# Summary

This paper introduces a novel method (CRC) to unify active learning (AL) and semi-supervised learning (SSL). The paper claims that designing labeled datasets by querying can control the convergence rate. To design queries, the proposed method selects unlabeled data points that maximize the smallest eigenvalue of Neural Tangent Kernel (NTK). Empirical results are presented.

# Strengths

1. I believe combining AL and SSL is an important research direction.
1. This paper shows a novel use of NTK that has many potential use cases.

# Weaknesses

1. The authors say existing approaches that combine SSL and AL (e.g., MMA by Song+2019) are "limited to independent combinations of SL and SSL." However, both MMA and CRC alternatively use SSL and AL.
1. The authors say the performance of baseline (FixMatch, an SSL method) is upper bound because the model is trained longer (from 3x to 5x more than this paper) and labeled data are balanced.
  - Can CRC achieve comparable performance to FixMatch if trained longer? Or is performance degeneration mostly due to NTK?
  - I think AL methods can ideally query and obtain more useful labels of support vectors between difficult boundaries compared to SSL methods. Thus, if the same number of labels is available, shouldn't SSL be the lower bound?
1. To use NTK, the method needs to train the model from random states every time labels are updated and thus, needs extra computational resource.
1.  Queried labels depend on the current model developed from certain initial states. However, after querying, the model is reinitialized. I guess this degenerates performance.
1. Despite the interesting concept, the empirical results are not appealing.

# Feedback and Questions

* Why some values in tables are with std and others are not? (e.g., Table 2) Also, a value in table3 for confidence in labeled images of 400 is missing.
* $C$ on page 3 is not defined.

---

> ### Author Response · Authors · 2020-11-19
> **Response**
>
> Thank you for your feedback!
> * Limited to independent combinations of SL and SSL
>     * We interpret the reviewer’s comment of “SL and SSL” as a typo of “AL to SSL” as stated in the manuscript and respond to the latter. We are aware that both our setting and MMA considers the setting of alternating AL and SSL, and we were referring to the fact that their AL algorithm (diff2 and “max” which we refer to as “confidence”, with enforced diversity) is not designed considering how the model is trained using SSL. In contrast, CRC is based on the assumption that SSL provides a nice target function (f-infinity) and seeks to estimate the training dynamics as a combination of the labeled set and candidate batches to-be-queried. We also suggest in the conclusion section how SSL can incorporate the information provided by CRC (time-horizon), or more generally, SSL algorithms can be designed to consider the AL algorithm used to construct the training set.
> * Performance degeneration
>     * All baselines and CRC use FixMatch as the SSL algorithm, and only influence how the labeled set is constructed. Therefore, the performance generation is not due to NTK.
>     * We terminated training when either the number of epochs reached 350 or the performance did not improve for 50 epochs, with the latter occurring more often. Thus, we believe that the performance degeneration is not resulting from the smaller number of iterations.
>     * There are several factors that may have affected the performance degeneration of FixMatch. One issue is addressed in the overall response, and our additional experiments using different implementations and SSL algorithms partially resolve this issue. Additional responses specific to the reviewer:
>
>         * We are claiming that the difference in SSL and our ASSL setting (not algorithm) puts performances reported in typical SSL settings to an advantage. Typical SSL settings construct the labeled set by randomly sampling the same number of images per class. It is well known that class-imbalance adversely affects the resulting performance, and SSL algorithms robust across various levels of class-imbalance have not yet been developed to the best of our knowledge. However, it is impractical to assume that an AL algorithm can query samples such that the resulting class distribution is uniform, as knowledge of unlabeled images’ class prior to querying would defeat the purpose of obtaining their labels. Our performance reports on “random” is a good reference point on how FixMatch performs in the presence of class-imbalance.
>         * Given our setting, AL algorithms should ideally improve upon “random” sampling as the reviewer noticed, and our experiments confirm that Entropy and CRC do outperform “random” sampling in most cases.
>         * Considering these responses above, we added in Section 4.3 the average performances of all implementations when using balanced sampling (i.e. SSL setting).
>
>     * Obtain more useful labels of support vectors between difficult boundaries
>         * Perhaps surprisingly, a margin-based approach to active learning (see reference) is not necessarily the best method to select samples in the supervised learning setting.
>         * Reference: “Adversarial Active Learning for Deep Networks: a Margin Based Approach “

---

> ### Author Response · Authors · 2020-11-19
> **Continued Response**
>
> * Need to train model from random states on each query step
>     * Neither CRC nor baseline algorithms require that the model is trained from random states between each query step. Between choosing to (1) train a model from scratch every time the labeled set is updated and (2) continually training the model obtained in the previous step, we chose to validate the performance of AL algorithms using (1) because of its reduced degrees of freedom which we believe would be a more accurate comparison of AL algorithms. All experiments may as well have been performed by continually training the model between dataset updates.
>     * An example of configurations to consider when continually training is the setting considered in MMA. The authors trained a model for 262,144 steps on the initial labeled pool, and continued to train the model for 32,768 steps after obtaining more samples. However, different AL algorithms may have different optimal configurations. For example a model trained on a dataset constructed using entropy (or any arbitrary AL algorithm) may benefit from training more than 50,000 steps between each dataset updates, whereas the model trained on a randomly sampled dataset may converge within 32,768 steps.
>     * In addition to the reduced degrees of freedom, training from scratch at each dataset updates comes with additional observations of interest, namely, transferability of the constructed dataset. For an AL algorithm to be practical, a dataset of 300 labeled samples constructed using some AL algorithm equipped with a model trained on 250 labeled samples should also be useful in training another randomly initialized model (or even a different architecture). If however, only the model trained on the 250 labeled samples used to construct the 300 labeled samples can perform well and other models fail to perform well on the same 300 labeled dataset, the AL algorithm would be deemed inefficient, as the dataset with 300 labeled samples is useful in training only a single model whereas one may hope to train different models on the same dataset (e.g. for ensembling).
> * Feedback and Questions
>     * All values have been updated, with C defined as the number of classes.
>
> We believe the above responses cover all the reviewer's comments. Please let us know if there is anything we missed.

---

### Official Review · AnonReviewer4 · 2020-10-28
**An active learning based strategy to improve the convergence rate for semi-supervised learning inspired by the neural tangent kernel**

**Rating:** 6
**Confidence:** 3

**Review:**

This paper proposed an active learning strategy to improve the convergence rate for the semi-supervised deep learning algorithm. When the SSL objective could learn a good approximation of the optimal model, the proposed method efficiently converges to the result with a few queries. The essential technique used here is the recent advance of the neural tangent kernel; that is, when the eigenvalue of the neural tangent kernel is large, the convergence rate is in turn fast. Inspired by this theoretical results, the authors provided the learning algorithm to maximize the smallest eigenvalue of the neural tangent kernel. Empirical studies show the effectiveness of the proposed algorithm.



pros
1. The authors provide an interesting idea of using the theoretical results of the neural tangent kernel to improve the convergence rate by actively labeling the unlabeled data in semi-supervised learning. Such a design is simple but efficient.
2. Compared with previous works, the proposed method also considers the influence of labeled data (calculate the eigenvalues jointly with labeled data) and uses the information provided by the SSL objective.

cons
1. This paper provides an insight into choosing the unlabeled data to improve the convergence rate of SSL methods. Empirical evidence shows the effectiveness of the proposed algorithm, but there lacks solid theoretical support.
2. The improvement of the proposed algorithm over the baseline methods (query by entropy) is limited. As shown in Table 3, the proposed CRC is not the best choice in some cases. There lack the explanation of this phenomenon.

---

> ### Author Response · Authors · 2020-11-19
> **Response**
>
> Thank you for your feedback!
> * Lack of solid theoretical support
>     * We theoretically prove in Section 3.2 some conditions under which CRC, which uses the empirical NTK instead of the true NTK, can still improve the rate of convergence (which depends on the unknown, true NTK’s minimum eigenvalue), by estimating a tractable bound in terms of the empirical NTK (which is computed when performing CRC).
> * Limited performance improvement over baseline methods, with not being best in some cases.
>     * While CRC is admittedly not the best across all cases, it outperforms all baseline algorithms in most cases, and this is further evident in the additional experiments reported in our updated manuscript.
>     * Unfortunately, existing DL-based AL algorithms have been unable to outperform random sampling in the supervised learning setting by a margin noticeably larger than that obtained by CRC. However, we believe that providing a new approach to AL (and its connections to modern theory of DL) can serve as a strong reference for the still-maturing field.
>     * On the other hand, most AL experiments have been conducted using small networks (e.g. LeNet, VGG, or custom shallow networks), but our experiments consider a significantly larger, modern architecture (WRN).

---

> > ### Comment · AnonReviewer4 · 2020-11-24
> > **Response to authors**
> >
> > Thanks to the authors for the detailed response. I think the theoretical analysis makes this manuscript more self-contain after the revision. The empirical results are limited but show the evidence to be beneficial. I keep my score as 6.

---

### Official Review · AnonReviewer2 · 2020-10-29
**A new active semi-supervised learning algorithm based on NTK theory**

**Rating:** 5
**Confidence:** 4

**Review:**

This paper makes an attempt at combining semi-supervised and active learning. The authors note that in restricted settings, SSL and AL can achieve exponential improvements over standard supervised learning with random sampling. Instead, this work attempts to use active learning to speed up the convergence to the asymptotic classifier (in terms of epochs) and semi-supervised learning to achieve the exponential improvement in data efficiency. This work is inspired by a neural tangent kernel analysis.

Strengths:
 - Integrating active learning into SSL techniques that perform well for image datasets is a good goal, as SSL techniques have dramatically improved for image datasets in the recent past.
 - Promising empirical performance against a few other algorithms.

Weaknesses:
 - The empirical results are only reported for a single dataset (CIFAR10).
 - Although the proposed algorithm, Convergence Rate Control (CRC), is supposed to speed up convergence, only final accuracy is reported. Thus, the reason for the increase in performance is not validated.
 - The theory is perhaps limited because of crude approximations, which aren't validated in terms of approximation error, but overall performance (see table 1 and 2). In particular, the algorithm uses only using the gradient from the final layer parameters, a standard batch approximation, and a "group" approximation.

Questions:
 - Can the authors describe more the "ill-posed nature" of active learning?



After reading author response ================================

Thank you for the response.

It's good to know that CRC outperforms random sampling on a couple other image datasets, but would be informative to see the results with respect to other AL techniques.

I wasn't quite able to follow the logic of section 3.2, but this seems on the right track.

The issue I see with using the final layer is not performance related, it's that it seems to throw out the theory you claim to be using. For instance, the fact that the performance dramatically improves is troubling and makes me wonder if the algorithm is working because of the theory or for a different reason. On the other hand, if you aren't able to make the theoretical justification more compelling, I think it would be fine to say the theory is just for inspiration.

I agree there is no batch approximation when G=Q. I might be missing something, but the paper says "All experiments hereon use G = Q/10 considering the speed vs. performance trade-off".

---

> ### Author Response · Authors · 2020-11-19
> **Response**
>
> Thank you for your feedback!
> * Empirical results for a single dataset:
>     * We have updated the manuscript to include experiments comparing the proposed algorithm with random sampling on two additional datasets: SVHN and CIFAR-100. Due to time constraints, we were unable to compare with other AL algorithms for these experiments.
> * Reason for increase in performance is not validated and limited theory
>     * In consideration of limited theory and performance validation, we added section 3.2 that theoretically proves CRC improves the true convergence rate up to an additive factor. The analysis describes how CRC implicitly allows larger learning rates (larger steps were observed to be taken for SSL algorithms) and its applicability to computing a tractable optimal step-size along with convergence rate in training DNNs.
> * Crude approximations using final layer’s gradients, “standard” batch approximation, and a “group” approximation
>     * Computing the empirical NTK using gradients over all layers requires orders of magnitude more computation in modern network architectures. An obvious alternative is to use the final layer’s gradients as we have done, but this may deviate significantly from the inspirational theory. We related this final layer-approximation with recent observations suggesting why using the final layer inherits its motivation of using all layers. Moreover, we confirmed in Table 1 that using the final layer’s gradients are sufficient to maintain the performance enhancements achieved by computing the empirical NTK over all layers’ gradients.
>     * We have further updated Table 1 to include more CRC experiments using all layers to further validate our claim for a final layer approximation.
>     * Batch and group approximation - we interpret the reviewer’s comments of “standard batch approximation” as approximating the influence of a batch using greedy approximations (coreset, BatchBALD). In this perspective, we respectfully disagree that CRC uses any kind of “standard batch approximation”, where G=Q computes the influence of candidate batches exactly (other than the final vs. all layer above).
>     * As for the group size approximation, this approximation comes into play because of a randomization over the search space, considering only |X_U|/G groups instead of the full search space with |X_U|-choose-Q candidate batches. The substantially reduced search space to |X_U|/Q candidates could potentially exclude optimal batches and compromise performance, for which we address by introducing the group-size and search over a slightly larger search space (by a constant multiplicative factor) which reduces the sub-optimality resulting from randomized search but instead computes the collective influence of a “group” smaller than the “batch” that is to be queried. Table 2 validates that smaller group sizes are consistently better than compromising the search space to quantify the collective influence of the full batch (G=Q). This table is also updated in the revised script to consider G=Q (exact batch computation, randomized search space) and more query steps.
> * Ill-posed nature of active learning
>     * We thank the reviewer for pointing this out. This is unintentionally misleading, and was supposed to refer to the sentence “This ambiguous characterization of how much information a sample’s label carries” later in the paragraph. We rephrased this paragraph accordingly.

---

### Author Response · Authors · 2020-11-19
**Overall Response to All Reviewers**

We thank all the reviewers for taking their time to read the manuscript and provide constructive comments which helped revise our work. All reviewers seem to appreciate the proposed active learning algorithm, its relation to the neural tangent kernel, and importance of combining AL with SSL.

Most concerns raised by the reviewers are addressed in the point-by-point responses and the revised manuscript, and we hope to clarify a few misunderstandings. In particular, we have extended most tables to include more query steps, added FixMatch experiment on CIFAR-100 with a different publicly-available PyTorch implementation, and Mean Teacher as the SSL algorithm for SVHN. We also rephrased parts of the manuscript to fix typos, rephrase misleading sentences, and include details on additional experiments, further described in the newly attached Appendix. The largest text-modification is an additional subsection with a theoretical analysis due to approximating the dynamics using the empirical NTK when performing CRC queries, in response to reviewers 1 and 2 (AnonReviewers 2 and 4) concerning theoretical support and crude approximations. In short, the analysis suggests a reason for CRC’s performance gains despite its approximation of the training dynamics.

Regarding the lower performance of the CIFAR-10 FixMatch implementation we used, there has been a very recent update (Nov. 2020) that has come to our attention, stating that the previous implementation had an issue with exponential moving average which resulted in performance degradation. However, this performance degradation is consistent across all AL algorithms and does not put CRC to any advantage. Furthermore, our additional experimental reports using a different implementation on CIFAR100 and a different SSL algorithm (mean teacher, again a third implementation) shows that CRC consistently outperforms random sampling. All SSL implementations were selected based on the best performing, publicly-available sources implemented in PyTorch at the time of our experiments. The average performances under respective implementations are now described in the updated manuscript.

---

### Decision · Program_Chairs · 2021-01-07
**Final Decision**

**Decision:**

Reject

**Comment:**

The paper investigates an active learning strategy for speeding up the convergence for SSL deep learning algorithms. When the SSL objective could learn a good approximation of the optimal model, the proposed method efficiently converges to the result with a few queries. The main idea is that when the eigenvalues of the NTK are large, the convergence rate is faster. The proposed algorithm maximizes the smallest eigenvalue of the NTK. An empirical investigation is also reported.
The reviewers appreciated the general idea, but questioned about the actual execution of this paper in terms of both experimental comparison and (lack of) supportive theoretical results. I would like to encourage the authors to consider improving their paper along one of these two lines.  Unfortunately, as it currently stands, this paper is not ready for publication.